# The Influence of Biological Factors on Haematological Values in Wild Marsh Harrier (*Circus aeruginosus*) Nestlings

**DOI:** 10.3390/ani11092539

**Published:** 2021-08-29

**Authors:** Urszula Zaremba, Zbigniew Kasprzykowski, Elżbieta Kondera

**Affiliations:** Faculty of Exact and Natural Sciences, Siedlce University of Natural Sciences and Humanities, Prusa 14, 08-110 Siedlce, Poland; zbigniew.kasprzykowski@uph.edu.pl (Z.K.); elzbieta.kondera@uph.edu.pl (E.K.)

**Keywords:** physiological status, nestling condition, hatching order, brood reduction, reference values, raptor, blood chemistry, haematology

## Abstract

**Simple Summary:**

Haematological examination can be used to address the health state of birds. The present study aimed to investigate how hatching order, brood size, and nest initiation date influence the haemoglobin level, red blood cell count, white blood cell count, and glucose level in the blood of raptor nestlings in which brood reduction occurs during nestling rearing. Blood samples were collected to assess the health status of wild marsh harrier nestlings in the agricultural landscape of eastern Poland. Statistical analyses revealed that hatching order, but not brood size or nest initiation date, had an impact on the haematological data of nestlings. Hatching order affected these data in that haemoglobin levels and red blood cell counts gradually decreased, whereas glucose levels and white blood cell counts increased, from the first- to the last-hatched nestlings. The poor condition of the youngest nestlings reflected in these indices may increase the likelihood of their perishing, and consequently, of brood reduction.

**Abstract:**

Marsh harrier (*Circus aeruginosus*) is a species with obligatory cainism, in which hatching asynchrony creates a pronounced size hierarchy in nestlings. The size-related competitive advantage of older nestlings means that they tend to dominate the younger ones, and brood reduction occurs in most nests. The aim of the study was to reference values and carry out a haematological examination in order to evaluate the physiological status and health of nestlings with respect to hatching order, brood size, and nest initiation date. To do so, we examined 19 nests with a total of 58 nestlings from a free-living population of this species located in fishpond complexes in the agricultural landscape of eastern Poland. Repeated blood samples (118 in all) were collected from nestlings. The following parameters were measured using fresh full blood: red blood cell count (RBC), haemoglobin level (Hb), white blood cell count (WBC), and plasma glucose level (Glu). The data were analysed using generalized linear mixed models and linear mixed models ((G)LMM). The study revealed that hatching order, but not brood size or nest initiation date, affected the physical condition of marsh harrier nestlings. Hb levels and RBC counts gradually decreased, whereas Glu levels and WBC counts increased from the first- to the last-hatched nestlings. This result points to the generally poorer condition of the youngest nestlings compared with their older siblings. The poor physiological condition of the youngest nestlings may consequently increase the likelihood of their perishing, and hence, of brood reduction.

## 1. Introduction

In recent years, haematological examination has become a valuable tool in veterinary care and wildlife conservation for evaluating physiological status and health in birds [1,2]. Routine haematological analyses include the evaluation of blood cell counts and other cell-related parameters, as well as measurements of such biochemical parameters as the concentrations and activities of plasma compounds. Haematological and plasma compound indices provide extensive information about bird oxygen transport capacity, immune potential, stress level, disease, intoxication, and nutritional status, and help to evaluate a bird’s overall physical condition [3,4,5].

To date, most studies examining the haematological and biochemical parameters of peripheral blood in raptors have been conducted on birds living in captivity [6,7,8,9]. It is therefore important to report reference values for wild individuals so that such data can be used as a reference for health diagnosis in clinical practice. However, there are several factors that can affect haematological values in the blood of raptors. These values can differ not only between species but also within the same species. An increasing number of studies have examined factors that could influence variation in the haematological data of raptors within the same species. For example, values reported in the literature indicate that haematological parameters in raptors can vary with age [10,11,12], sex [13], brood size [14], reproduction, and moult [15]. However, the majority of these studies were conducted on captive birds at rehabilitation centres, and studies of free-living populations of such species are still scarce [10,14,16,17].

Earlier studies of marsh harrier (*Circus aeruginosus*) haematology and blood chemistry involved both captive adult birds [18] and fledglings [19]. However, previous haematological data obtained from fledglings reported only reference values for haematocrit (Ht), total protein, and glucose (Glu). The aim of this study was to report a representative haematological range for haemoglobin (Hb) and Glu levels, as well as red blood cell (RBC) and white blood cell (WBC) counts, in free-living nestlings of this species nesting in fishpond complexes in the agricultural landscape of eastern Poland. Previous studies of the genus *Circus* (Harriers) investigated the influence of factors such as sex [18] and brood size [19] on haematological values. However, none of those studies examined hatching order or the first-egg laying date (FED) as factors that might influence haematological values in nestlings. Therefore, besides reporting reference values, the main objective of the present study was to evaluate whether Hb and Glu levels, as well as RBC and WBC counts, in the blood of free-living marsh harrier nestlings varied with biological factors (i.e. hatching order, brood size and first-egg laying date (FED)). 

Marsh harrier is a ground-nesting raptor which builds its nest in an aquatic environment. The nestlings hatch asynchronously, and the egg hatching period can last for up to seven days, leading to a visible size difference between the older and younger nestlings. The size-related competitive advantage of older nestlings means that they tend to dominate the younger ones. Marsh harrier is a species with obligatory cainism, so brood reduction occurs in most nests as a result of siblicide, and usually only three nestlings fledge from a complete clutch of five eggs [20]. We thus hypothesized that hatching order would have the biggest influence on the condition of individual nestlings, as reflected by their haematological parameters. We anticipated that last-hatched nestlings, in particular, being at a competitive disadvantage, would be in a worse condition than the first-hatched ones. We also expected that the number of nestlings would affect their blood parameters, especially in larger broods, and that the blood chemistry would vary with the nest initiation date.

## 2. Materials and Methods

### 2.1. Study Area

The study was conducted during one breeding season, in 2019. The study sites were located in the agricultural landscape of eastern Poland on four fishpond complexes—Siedlce, Rudka, Szostek, and Moscibrody (52°05′–52°11′ N, 21°58′–22°18′ E)—all of which are used mainly for the commercial breeding of common carp (*Cyprinus carpio*). The ponds varied in area from 65 to 203 ha, and the total area was 443 ha. They were all situated within 20 km of each other. Most of the ponds were partially covered by tall marsh vegetation consisting of common reedmace (*Typha latifolia*), common reed (*Phragmites australis*), and sedges (*Carex* spp.). These plants tend to proliferate rapidly and can quickly cover the surface of a pond or wetland, creating a suitable breeding habitat for marsh harrier. The study sites were managed extensively with occasional reed cutting. The depths of the ponds were similar, but water levels within the emergent vegetation fell gradually as the breeding season progressed.

### 2.2. Field Procedures 

To locate active nests, each of the four ponds was visited at 1-to-3-day intervals between mid-April and mid-May. The birds were observed carrying nest material to the emergent vegetation belt and during aerial food-passes near their potential nest site. Having selected a potentially favourable site, the observers inspected the vegetation belt on foot along fixed line transects. Once located, the nests were numbered and their positions recorded on a GPS device. Upon discovering a nest that had already been initiated, the first-egg laying date (FED) was calculated by counting back on the assumption that a female lays one egg every two days. A total of 19 nests in the study area were visited at 5-to-7-day intervals during the nestling period and a total of 118 blood samples were taken from 58 nestlings that were in the first, second, and third week of life. We analysed three nests with only one nestling, two nests with two nestlings, five nests with three nestlings, and nine nests with four nestlings. In the majority of nests that initially contained five nestlings, the last-hatched ones perished soon after hatching; the brood size had thus been reduced even before we were able to take blood samples. In such cases, the nest was categorized as having a brood size of the four surviving nestlings. After hatching, the chicks were temporarily marked with coloured plastic rings; these were subsequently removed just before fledging. Samples were taken between 11:00 and 16:00 hrs in order to minimize variations in plasma analytes due to circadian rhythms. During each visit, the numbers of nestlings were noted, and a blood sample was taken from each one from the brachial wing vein on the underside of its wing. Blood was taken from each nestling at the nest on average from 1 to 3 times. The number of blood collection repeats varied because some of the nests were depredated at an early stage of nestling rearing, and some nestlings perished as a result of brood reduction or predation. To take the blood samples, a single puncture with a sterile needle was made in the brachial wing vein, and 75mm/65 μl heparinized haematocrit capillaries (Hirschmann Laborgeräte, Eberstadt, Germany) were used to suck the blood into the tube via capillary action. A volume of 0.2 ml of blood was taken from each nestling and then transferred to a sterile 1.5 ml plastic Eppendorf tube. After sampling, the blood flow was stemmed by applying pressure to the puncture with a piece of tissue paper. The procedure itself took up to 5 minutes at one nest. The blood was collected directly in the field, during the nest inspections. This reduced the level of stress experienced by the chicks; this was much lower compared with that caused by the transport of birds to the laboratory and their subsequent return to the nest. Immediately after collection, the blood was kept chilled in a portable refrigerator until it could be delivered to the laboratory. 

### 2.3. Sample Analyses

Following delivery of the blood samples to the laboratory (about 1 hour later), basic haematological parameters were determined using standard methods. The following parameters were measured using fresh full blood: erythrocyte count (RBC), haemoglobin level (Hb), leukocyte count (WBC), and plasma glucose (Glu) level. The haemoglobin level was measured spectrophotometrically at wavelength 540 nm after mixing the blood 1:10 with Drabkin’s solution and converting the haemoglobin to cyanmethaemoglobin. Hb values were then calculated from the relationship between dilutions of haemoglobin standard and their extinction. Erythrocyte (RBC) and leukocyte (WBC) counts were performed in Bürker haemocytometers (Paul Marienfeld GmbH & Co. KG, Lauda-Königshofen, Germany) in blood diluted 100 times with Hayem’s solution. The glucose level was measured using an Accu Check (Roche, Basel, Switzerland) glucometer.

### 2.4. Statistical Analyses

All the statistical analyses were performed using R-studio [21]. We focused our analysis on investigating how hatching order (labelled a, b, c, and d), number of nestlings (labelled 1, 2, 3, and 4), and first-egg laying date (FED) affected RBC and WBC counts, as well as Hb and Glu levels in marsh harrier nestlings. General linear mixed models (GLMM) and linear mixed models (LMM) models were applied to calculate the data. Hatching order, brood size, and first-egg laying date (FED) were treated as fixed factors, while nest identity, blood collection date, and nestling identity were introduced as random factors. Hatching order and number of nestlings were formatted as factorial variables, and first-egg laying date as a numerical variable. Model selection was performed using Akaike’s Information Criterion (AICc) [22]. The models with interaction terms between the predictors performed no better than the models which included only additive parameters in the fixed component of G(LMM). The global model was thus: X ~ hatch.order + FED + brood size + (1|nest) + (1|blood collection) + (1|nestling). For the Glu model, the dependent variable followed the normal distribution according to Shapiro–Wilk’s normality test (*p* = 0.943), so the LMM model was applied. Of the five candidate models differing in the fixed component of variables, the one including hatch order and brood size was considered the best (AICc = 1081.48; global model AICc = 1082.82). In the Hb, RBC, and WBC models, the dependent variable was not distributed normally (Shapiro–Wilk test; *p* = 0.001, *p* < 0.001, *p* < 0.001, respectively). Therefore, GLMM models with gamma distribution and link log function were applied to these parameters. Of the five candidate models for the Hb model, the one with hatching order as the only predictor was considered the best (AICc = 1272.92; global model AICc = 1275.73). From the set of five candidate models for the RBC model, the one with hatching order as the only predictor performed the best (AICc = 210.13; global model AICc = 215.60). In contrast, of the five candidate models for the WBC model, that with the lowest AICc score (1074.05) was the one with hatching order and brood size as predictors (global model AICc = 1076.05).

## 3. Results

Fifty-eight free-living marsh harrier nestlings were studied. Reference values of selected haematological parameters (RBC, WBC, Hb) and blood Glu levels are shown (Table 1). The number of blood samples varied between blood collections because some of the nests were depredated at an early stage of nestling rearing, and some nestlings perished as a result of brood reduction or predation. The LMM models showed that blood Hb levels gradually decreased with hatching order (Figure 1) and that they were the lowest for the last, marginal nestlings. However, hatching order significantly affected blood Hb levels only in the last-hatched chicks (*p* = 0.031; Table 2). Brood size and first-egg laying date were not introduced into the final model as these predictors did not influence the Hb level in the nestlings’ blood.

The pattern was similar for the RBC count. The best model selected revealed that only hatching order had an impact on nestling blood RBC. As in the Hb model, the value of this factor declined gradually with hatching order (Figure 1). The decrease was also significant only for the last, marginal chicks (*p* = 0.038; Table 2). Brood size and first-egg laying date did not influence the Hb level in the nestlings’ blood. In contrast to RBC and Hb, Glu levels rose from the first-hatched until the last-hatched nestling. Glu in the blood of nestlings was significantly higher for the third (*p* = 0.048) and youngest nestlings (*p* = 0.017; Figure 2). Although the final Glu model included brood size, this predictor had no significant influence on the Glu level in the nestlings’ blood. In the WBC model, hatching order and brood size were included in the final model, but only hatching order significantly influenced the WBC count in the nestlings’ blood (Table 3). WBC increased gradually with hatching order, peaking in the last-hatched chicks (*p* < 0.001; Figure 2). The first-egg laying date received no support from the Glu and WBC models.

## 4. Discussion

The reference Glu values obtained for marsh harrier nestlings did not differ from those obtained for free-living marsh harrier fledglings in Spain [19] and were nearly similar during all three study trials. These values were lower than those reported for free-living nestlings in the *Accipitridae* family [10,11,23,24,25], except booted eagle (*Hieraaetus pennatus*) [26]. The plasma Glu level in the blood of marsh harrier chicks was also lower than in a number of European Strigiformes, but not barn owl (*Tyto alba*) [27]. The RBC counts of marsh harrier nestlings were similar to those obtained from free-living fledglings of Montagu’s harrier (*Circus pygargus*) [14] and peregrine falcon (*Falco peregrinus*) [28]. However, the RBC count was lower than in adult captive marsh harriers [18]. A similar pattern prevailed in young and adult bearded vultures (*Gypaetus barbatus*) [11]. Although the WBC counts in marsh harrier nestlings were high during the first blood collection, they decreased in the last samples, reaching values similar to those in adult captive marsh harriers [18]. These values were also slightly higher than those reported for Montagu’s harrier fledglings [14] but smaller than those reported in peregrine falcon nestlings of a similar age [28]. The blood haemoglobin levels in marsh harrier nestlings were nearly the same as those obtained from adult captive marsh harriers [18] and fledglings of Montagu’s harrier [14], but smaller than in peregrine falcon nestlings [28].

Our study shows that hatching order affects the blood haematology of marsh harrier nestlings. Above all, nestling age negatively influenced Hb levels and RBC counts. These parameters decreased with hatching order, with the decrease being the most pronounced in the youngest nestlings (Hb and RBC). Low RBC and Hb values in younger chicks were found earlier in species with asynchronous hatching and brood reduction, like white stork (*Ciconia ciconia)* [29]. Moreover, haematocrit levels were low and body condition poor in the youngest nestlings from asynchronously hatched little egret (*Egretta garzetta*) chicks [30]. Marsh harriers typically lay five eggs, but the resources provided by the parents may not always be sufficient to rear the whole brood successfully. Brood reduction occurs in most nests, often regardless of the size of the brood. As a result of siblicide, usually only three nestlings fledge from a full clutch of five eggs [20]. In our study, the significantly lower RBC counts and Hb levels in the youngest nestlings may reflect a competitive disadvantage compared to their older siblings. It is known that older marsh harrier nestlings tend to dominate the younger ones and prevent them from consuming food brought by the parents. It has been demonstrated that, in general, nestlings reared in a habitat with poor food resources have lower blood Hb levels than their counterparts reared in a food-rich environment [31]. Hence, we may suspect that the last-hatched marsh harrier nestlings in our study may, as a result of intra-brood competition, not have acquired sufficient nutrients to maintain higher blood Hb levels and RBC counts. This potential decrease in food intake may have led to their inferior physical condition compared to the earlier hatched nestlings and thus, to the lower Hb levels and RBC counts. The Hb level has been shown to be a general indicator of individual physiological condition, being positively correlated with breeding and fledging success [32,33,34,35]. Moreover, low Hb levels and RBC counts generally indicate poor health and anaemia [3,5], which can impair the ability of the blood to carry oxygen to the tissues. Although we did not evaluate blood parasites in our study, it is worth noting that a lower Hb level may also be a sign of the adverse impact of blood parasites, which may likewise reduce the chances of survival in juvenile altricial birds [36]. We therefore conclude that low RBC counts and Hb levels in last-hatched marsh harrier nestlings are a sign of their poor physical condition and increase the likelihood of their perishing and of brood reduction.

While RBC counts and Hb levels decreased with nestling hatching order, the opposite occurred in the case of blood Glu levels, which were highest in the third and fourth nestlings. The Glu level in blood can provide information about the nutritional state of an individual. García-Rodríguez et al. [37] found that blood Glu levels were elevated in individuals that had suffered prolonged starvation. In addition, Montoya et al. [38] stated that baseline Glu was negatively associated with survival, increasing in response to stress and adverse environmental conditions. As with the RBC count and Hb level, this could mean that the food intake is not equal across the nestmates and that the first-hatched nestlings, enjoying a competitive advantage because of their larger size, potentially dominate their later-hatched siblings, excluding them from the food resources provided by parents. This could result in periods of starvation in younger nestlings and elevate their blood Glu levels. 

The WBC count also differed according to hatching order in marsh harrier nestlings. Generally, an elevated WBC count could be an indicator of the good condition and health of the immune system [39]. However, since we recorded decreases in Hb levels and RBC counts along with elevated Glu levels in the blood of later-hatched nestlings, all of which are indicative of poorer health, we suspect that high WBC counts in the youngest chicks could be a symptom of stress, inflammation, or infection [40,41]. Marsh harrier is a species with obligatory cainism, in which brood reduction occurs in most nests as a result of siblicide. Inflammation and stress responses in the youngest nestlings could be due to physical damage sustained in conflicts with older nestmates. These peck the younger ones on the head, possibly causing wounds and skin infections, which often result in the death of the attacked sibling or at least much reduce its chances of survival. Because hatching is asynchronous, there is a very pronounced size difference between the oldest and youngest nestlings in the first half of the nestling phase (up to 15 days old). Having a small body size, the youngest nestlings are often incapable of competing with their older siblings. The higher WBC count in the blood of the youngest nestlings could also be a sign of inflammation as a result of their greater susceptibility to parasites, previously linked with low Hb [42].

Interestingly, brood size had no influence on the haematological parameters of marsh harrier nestlings. This could be because the parents adjust the number of nestlings to their own rearing abilities and experience, thereby achieving an optimal brood size. However, owing to rapid brood reduction (the mortality of fifth nestlings in our study) ([43], in press), we did not have sufficient data to analyse the influence of all five brood categories on haematological values. Moreover, these values could differ in broods of five nestlings: Limiñana et al. [14] demonstrated that Hb levels were lower in broods of five in Montagu’s harrier and may have reflected the excessive cost of rearing such large broods. Further studies are needed to evaluate the influence of larger brood sizes in marsh harrier on haematological parameters. Furthermore, we failed to detect any influence of nest initiation date on blood chemistry values in marsh harrier nestlings, in contrast to some passerine species [44]. Because of the seasonal change in food resources, we expected the haematological parameters to differ between earlier- and later-hatched nestlings. However, it seems that the nest initiation date does not affect the physiological state of marsh harrier nestlings. The reason for such a pattern could be that nests initiated later in the season often contain fewer eggs, so nest-mate competition may well be less intense in such nests. We conclude that hatching order has the greatest influence on the physical condition of marsh harrier nestlings. High blood Glu levels and WBC counts, and low Hb levels and RBC counts in the youngest nestlings may indicate their overall poorer condition in comparison with their older nest mates and decrease their chances of survival. 

More studies are needed to evaluate stress levels in nestlings with respect to hatching order. For example, it would be useful to measure the ratio of heterophils to lymphocytes (H/L), which is a standard tool for assessing long-term stress in birds, including raptors [45]. In addition, stress-related high corticosterone levels have been reported in feathers of later-hatched nestlings of upland buzzard (*Buteo hemilasius*) [46]. Sibling competition also caused physiological developmental stress reflected in decreased body condition and increased allostatic load for younger lesser kestrel (*Falco naumanni*) nestlings [47]. By combining body condition indexes, blood examination, and ptilochronology, it may be possible to greatly expand current knowledge about the different stress levels experienced by siblings.

## 5. Conclusions

We have demonstrated that hatching order, but not brood size or nest initiation date, influences haematological values and plasma glucose levels in wild marsh harrier nestlings. However, owing to the rapid reduction of broods initially containing five nestlings, we were unable to detect the effect of all five brood categories on the Hb levels, RBC and WBC counts, and Glu levels in nestling blood. Hatching order affected these indices in that Hb levels and RBC counts in blood gradually decreased, whereas Glu levels and WBC counts increased from the first- to the last-hatched nestlings. Thus, we conclude that the poor health status of the youngest nestlings may be due to their inferior position in the sibling hierarchy and may increase the chances of their perishing and, hence, of brood reduction. Despite the small sample size of nests in our study, the reference haematological values and plasma Glu levels that we obtained may be useful for evaluating the health of marsh harrier nestlings in both ecological studies and veterinary practice. More studies are needed to evaluate stress levels among nestlings according to hatching order and brood size. It would be useful to measure the ratio of heterophils to lymphocytes (H/L), and also corticosterone levels, both parameters being associated with long-term stress. Investigating blood parasites in nestlings may also be helpful in expanding knowledge about the mechanisms governing cainism.

## Figures and Tables

**Figure 1 animals-11-02539-f001:**
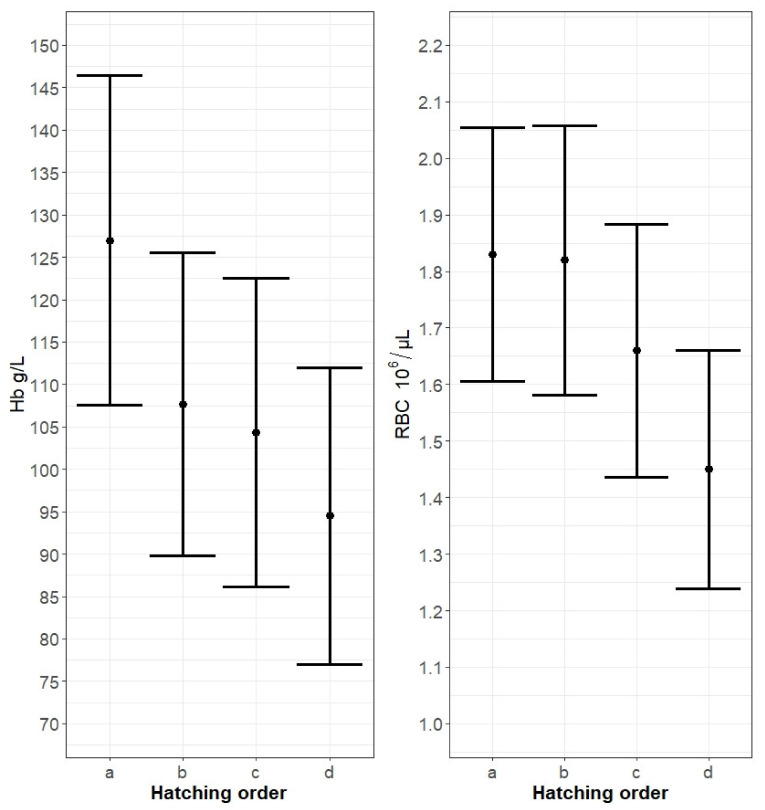
Variation of haemoglobin (Hb) level and red blood cell (RBC) count in the blood of free-living marsh harrier nestlings in relation to hatching order. The hatching order of nestlings is labelled chronologically with letters (**a**, **b**, **c**, and **d**). Modelled means ± standard error (SE) for the best model selected are shown. Values extracted using the lsmeans package.

**Figure 2 animals-11-02539-f002:**
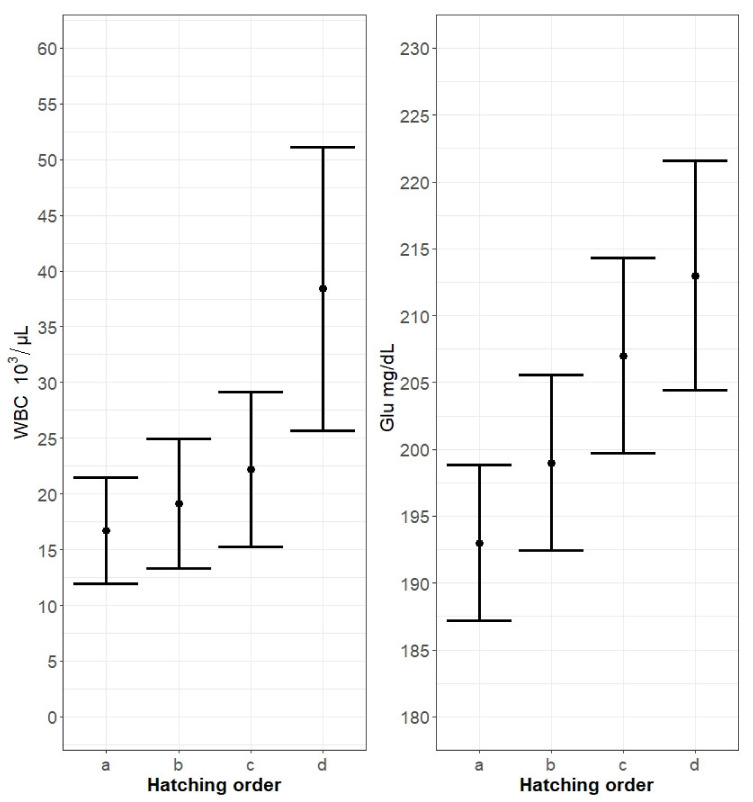
Variation of WBC (white blood cell) count and Glu (glucose) level in the blood of free-living marsh harrier nestlings in relation to hatching order. The hatching order of nestlings is labelled chronologically with letters (**a**, **b**, **c**, and **d**). Modelled means ± standard error (SE) for the best model selected are shown. Values extracted using the lsmeans package.

**Table 1 animals-11-02539-t001:** Descriptive statistics and reference values of selected haematological parameters: RBC (red blood cell) and WBC (white blood cell) counts, Hb (haemoglobin) and Glu (glucose) levels in blood of free-living marsh harrier nestlings.

Blood Parameter	Mean ± SD	Median	Minimum and Maximum Values
First blood collection (N = 58)			
RBC [10^6^/µL]	1.70 ± 0.77	1.61	0.49–5.02
WBC [10^3^/µL]	45.65 ± 37.41	31.79	0.33–173.80
Hb [g/L]	124.75 ± 66.42	115	30–430
Glu [mg/dL]	202.91 ± 25.15	205	152–262
Second blood collection (N = 46)			
RBC [10^6^/µL]	2.02 ± 0.66	2	0.60–3.67
WBC [10^3^/µL]	31.31 ± 25.79	19.79	1.04–98.87
Hb [g/L]	132.53 ± 50.14	120	44–250
Glu [mg/dL]	187.68 ± 39.68	195	26–280
Third blood collection (N = 14)			
RBC [10^6^/µL]	1.91 ± 0.86	1.63	1.18–3.98
WBC [10^3^/µL]	16.12 ± 10.46	13.15	5–39.58
Hb [g/L]	139.64 ± 77.27	117.5	25–290
Glu [mg/dL]	191.21 ± 26.82	184	161–254

**Table 2 animals-11-02539-t002:** Estimated model coefficients for the LMM model of Hb (haemoglobin) level and RBC (red blood cell) count in the blood of free-living marsh harrier nestlings. The hatching order of the nestlings is indicated by letters (**b**, **c**, and **d**). Standard errors (SE) and *p*-values are shown for the fixed effects. The values of the random effects are variance estimates and their standard deviations (SD).

	Hb			RBC		
Fixed Effects	Estimate	SE	*p*-Value	Estimate	SE	*p*-Value
Intercept	4.843	0.152	< 0.001	0.603	0.123	< 0.001
Hatching order b	–0.164	0.112	0.145	–0.003	0.092	0.973
Hatching order c	–0.196	0.118	0.095	–0.098	0.096	0.310
Hatching order d	–0.295	0.137	0.031	–0.230	0.111	0.038
Random effects	Variance	SD		Variance	SD	
Nestling Nest	0.028 0.045	0.167 0.212		0.018 0.017	0.134 0.131	
Blood collection	0.002	0.050		0.007	0.084	
Residual	0.102	0.319		0.083	0.288	

**Table 3 animals-11-02539-t003:** Estimated model coefficients for the GLMM model of WBC (white blood cell) count and the LLM model of the Glu (glucose) level in the blood of free-living marsh harrier nestlings. The hatching order of nestlings is labelled chronologically with letters (**b**, **c**, and **d**); brood size is indicated by numbers (**2, 3**, and **4**). Standard errors (SE) and *p*-values are shown for the fixed effects. The values of the random effects are variance estimates and their standard deviations (SD).

	WBC			Glu		
Fixed Effects	Estimate	SE	*p*-Value	Estimate	SE	*p*-Value
Intercept	2.326	0.460	< 0.001	205.528	11.549	< 0.001
Hatching order b	0.135	0.164	0.412	5.796	5.861	0.339
Hatching order c	0.283	0.179	0.114	13.435	6.541	0.048
Hatching order d	0.831	0.206	< 0.001	19.616	7.775	0.017
Brood size 2	0.899	0.489	0.066	–23.003	13.717	0.099
Brood size 3	0.829	0.483	0.086	–7.713	13.308	0.564
Brood size 4	0.235	0.452	0.604	–18.998	12.565	0.137
Random effects	Variance	SD		Variance	SD	
Nestling	0.043	0.208		26.04	5.103	
Nest	0.171	0.414		88.98	9.433	
Blood collection	0.077	0.277		27.91	5.283	
Residual	0.351	0.593		538.49	23.205	

## Data Availability

Dataset supporting reported results can be found at https://figshare.com/s/153a99066f0cc34e4b91 (accessed on 25 July 2021).

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
