# Peer review of "The Influence of Biological Factors on Haematological Values in Wild Marsh Harrier (Circus aeruginosus) Nestlings"

_animals, 2021, doi:10.3390/ani11092539_

Round 1

Reviewer 1 Report

The paper presents the differentiation of erythrocyte (RBC) level, leukocyte level (WBC), hemoglobin concentration (Hb), and plasma glucose (Glu) in nestling of Marsh Harrier. It presents new data analyzed in the context of variation resulting from the order of birds hatching, brood size and hatching time. In recent years, haematological testing has become popular in assessing the physiological status and health of birds, but there is still little data on the variability of bird blood components and hematological parameters. Physiological studies of wild birds constitute an important source of data and the work submitted for review provides new information worthy of publication. Overall, it is well prepared - the manuscript needs some adjustments or clarifications, for which I suggest you below.

Introduction

Lines 43-55. I propose to shorten the text, not focusing on listing what are the parameters of red blood, white blood, etc., but rather briefly present the results of the research so far, which focus on the assessment of the physiological condition of birds / bird health through hematological parameters relevant to the presented work.

Lines 64-71. A very important element of the Introduction that sketches the basis for the research hypotheses and indicates the need to undertake research. He also proposes a slightly redrafting, based on important assumptions, where most of the physiological condition studies based on hematological studies concerned species where asynchronous hatching was not taken into account, that the previous research on marsh harrier, including chicks (see lines 82-83) did not take into account many hematological parameters and hence the purpose of the work and the hypotheses posed

Materials and Methods

Study area

Lines 93-96. Latin names Cyprinus carpio (line 93), Typha latifolia (line 95), Phragmites australis (line 96), Carex ssp. (Line 96) should be in italics and follow (Circus aeruginosus) in parentheses (see line 66)

Line 95 species name of Bulrush is Broadleaf Cattail Typha latifolia

Field procedure

Research procedures well described, sufficient for the scope of the work. Perhaps it is worth providing information about the age of the chicks (the age variation of the chicks) from which the blood was taken..

Sample analysis

Without comments

Statistical analysis

Lines 157-158 –  than the models without random components (AICc: 1049.71; -93.46; -45.01; 1119.33 respectively). I propose to remove - after all, choosing the GLMM or LMM data analysis method a piori, we introduce random factors; especially is imported including a random component as a date blood collection, because part of the variability is a covariance represent repeated measurements for a single subject

Lines 154-157 - Linear mixed model is a popular technique to deal with correlated data such as longitudinal data. The important problem of approximating is the precision of small sample. The authors' use of the corrected AICc value is absolutely justified, indicating awareness of the use of a small sample; but in this context there is no explanation as to how the degrees of freedom in the models were calculated - was it a model based on the Residual method, Satterthwaite-type approximation, containment method or approximation, e.g. by Kenward-Roger due to the sample size.

Lines 166; 169; 171, 175 - it is worth mentioning the results of testing the entire model next to the AICc value

Line 173 proposes to remove the word family; exponential distribution is a special case of gamma distribution

Results

Table 1

Label - Blood collection? What do the numbers 1,2,3,4 in this column mean? Do they match the next blood sample collection? If that were the case, in lines 109-111- the information about collecting blood from 57 chicks is given, for a total of 124 blood samples, in the Table 1 in the case of blood collection 1 - the number of samples is 58, and the total in Table is 122?

Figure 1 and 2 - no indication of the measurement unit on the Y axis; these are probably not data on RBC concentration in blood, but estimators of the equation ?? It can be noticed that the average RBC variability is at the level of about 0.13-0.26 on the Figure 1, and in Table 1 it is given at the level of 1-7-2.32 106/μL, also the scale of other parameters tested is different in Table 1 and in the figures. Wouldn't it be better to present the differentiation of physiological parameters in the measurement units used?

Figure 1-2 It does not seem justified to define the point of intersection of the Y axis as 0, after all, the scales of individual variables are not comparable with each other

Discussion

Lines 242-248 – it is worth noting that such a situation takes place regardless of the size of the brood

Line 255 -  the decrease in the concentration of RBC and HB takes place not only with infections with blood parasites, but also with other parasitoses; it is worth paying attention to exposure to toxins (eg, rodenticides, heavy metals), or the deficiency of elements, e.g. copper, affecting the production of red blood cells

Other

Line 319 – no information about the consent of the Local Ethical Committee for the conducted research

Reviewer 2 Report

The authors have used hematological data to evaluate the relative stress within a brood of Marsh Harriers. The study is well thought out,  implemented, and analyzed. It is, however (1) written in poor English, and (2) the discussion is very narrow-focused.   

In order to improve the quality and flow of the paper, the authors must have the paper proofread by a native English speaker. 

Also, in order to make the paper of interest to a wider audience, I recommend that the authors compare their findings with other studies that have evaluated the effects of stress on younger siblings in raptors by using other techniques - corticosterone levels, ptilochronology, etc. 

Title - delete "some"

Round 2

Reviewer 2 Report

The authors have made a good effort to improve the paper and this is evident in the changes they have made.

There are still some editorial glitches of language that need to be eased out by a professional editor.

Recommend the authors to change "bird of prey" with the word raptor.

Also, another paper relevant to your final part is Bakaloudis et al. 2020. J. Vert. Biol. 69:1-9. doi.org/10.25225/jvb.19058.
